# ENHANCING RAG WITH ACTIVE LEARNING ON CONVERSATION RECORDS: REJECT INCAPABLES AND ANSWER CAPABLES

## ABSTRACT

Retrieval-augmented generation (RAG) is a key technique for leveraging external knowledge and enhancing the factual accuracy of large language models (LLMs). However, RAG still faces challenges in ensuring fully reliable responses in all scenarios. To address this, it is essential to identify samples that tend to lead to unreliable outputs or guide LLMs toward factually correct responses, which experts then annotate to develop high-quality datasets for refining LLMs. However, the growing scarcity of such datasets makes their creation challenging. This paper proposes using the vast amount of conversations generated from widespread LLM usage to build these datasets, with the goal of training LLMs to appropriately handle queries outside its capabilities while providing accurate responses to manageable ones. Given the impracticality of having experts annotate all conversation records, we introduce AL4RAG, a framework that uses active learning to select the most suitable conversation samples for annotation, thereby optimizing model performance within a limited annotation budget. Additionally, recognizing that traditional active learning methods are not fully compatible with RAG due to unsuitable distance metrics, we develop a novel sample distance measurement for RAG active learning. Extensive experiments show that our method consistently outperforms baselines across multiple metrics.

## 1 INTRODUCTION

In recent years, large language models (LLMs) have demonstrated remarkable performance in diverse natural language processing (NLP) tasks, such as text classification (Abburi et al., 2023), summarization (Jin et al., 2024), and question answering (Zhuang et al., 2023). However, they frequently encounter the issue of output deviations (Huang et al., 2025), which undermines the reliability of their responses. Retrieval-Augmented Generation (RAG) (Lewis et al., 2020), integrated into leading models like GPT-4o (Hurst et al., 2024), Deepseek-v3 (Liu et al., 2024) and Claude-3.5 (Anthropic, 2024), aims to tackle this problem. RAG combines a retriever to fetch relevant documents and a generator to formulate answers, enhancing the model's reliability by leveraging external knowledge. Nevertheless, RAG cannot completely eliminate such deviations (Chen et al., 2024; Wood & Forbes, 2024), posing a persistent challenge in ensuring the quality of model-generated content.

The challenge of addressing output deviations highlights the need for systems that can effectively identify and manage situations prone to unreliable outputs. Our research aims to train models to reject queries outside its capability while ensuring stable and accurate responses to manageable ones, as illustrated in Figure 1(a). Achieving this goal requires high-quality model conversation datasets specifically designed to teach the model when to refuse to answer, while also including queries that the model can answer correctly in order to maintain its capabilities. However, relevant datasets are exceedingly scarce, and the creation of such datasets is highly challenging due to the need for extensive manual annotation, while the majority of existing model conversation data remains unlabeled. To address this, we propose leveraging active learning (AL) (Settles, 1995) to screen model conversation records. AL systematically identifies the most informative samples from extensive collections of unlabeled data, enabling the creation of a small but high-quality human-annotated dataset through focused manual annotation.

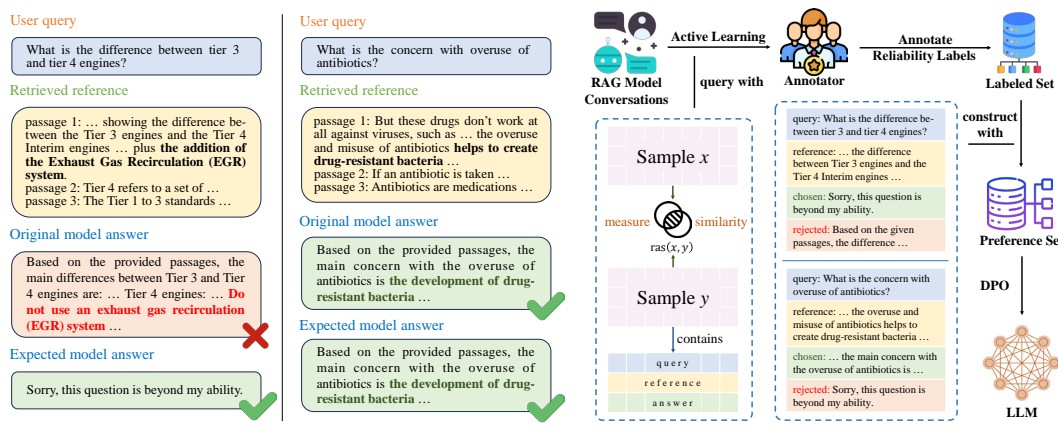

(a) An example related to our tasks                    (b) Overall framework

Figure 1: (a) An example regarding our tasks. In the scenario on the left, the original model provides an incorrect response to the user; in this case, we expect the model to decline answering the question, whereas in the scenario on the right, where the original model delivers a correct response, we aim for it to generate accurate responses more consistently. (b) Overall framework of our approach. The example of preference set construction is the same as (a).

Unfortunately, existing AL methods perform poorly in the RAG scenario. These methods can be categorized into two categories: uncertainty-based approaches that select samples with the model's prediction uncertainty, and diversity-based approaches that prioritize samples distinct from annotated ones to avoid redundancy. Traditional uncertainty-based methods demonstrate instability when applied to RAG-related tasks (Tsvigun et al., 2022; Snijders et al., 2023), while existing diversity-based methods (Maekawa et al., 2022; Xie et al., 2023) fail to consider the unique three-segment structure of RAG conversation records (e.g., a query, retrieved documents, and a model-generated response), resulting in insufficient diversity in the collected data. With the advent of LLMs, some researchers have employed LLMs to measure sample uncertainty (Li et al., 2024), but this approach requires LLMs to evaluate the uncertainty of all unlabeled samples sequentially, leading to excessive time and computational costs when applied in practical RAG scenarios. Therefore, developing AL methods tailored specifically for RAG systems becomes crucial.

To address this gap, this paper proposes **AL4RAG**, a novel AL strategy specifically designed for RAG models, as illustrated in Figure 1(b). We improve upon existing diversity-based methods (Tsvigun et al., 2022; Margatina et al., 2023) by independently considering the various fields of RAG data, which effectively accommodates the unique nature of RAG data. This approach selectively identifies the most diverse samples from unlabeled model conversation records. By annotating these strategically selected samples, we can construct a high-quality human-annotated preference dataset. However, to construct a preference dataset, it is essential to introduce a refusal option. Yet, in existing RAG model conversation datasets, such as RAGTruth (Niu et al., 2024), each question corresponds to only a single answer, which allows for the identification of unreliable outputs but lacks the option to mark when a model should refuse to answer. To address this challenge, we ask annotators to assign labels related to output reliability to the selected model conversations. Each sample is then labeled based on whether its output is reliable, indicating whether the model should prefer to answer or refuse to respond. This approach enables us to build a dedicated preference dataset. Using this dataset for model optimization can significantly improve the performance of RAG models, thereby achieving our intended objectives. In summary, our main contributions are:

- To the best of our knowledge, we are the first to propose an AL framework for RAG contexts. Our approach introduces an efficient selection strategy designed to address the distinctive data patterns inherent in RAG-based datasets.

- We proposed retrieval-augmented similarity (***ras***) for measuring the similarity between samples within the unique data patterns of RAG contexts, enabling more accurate measurement of sample distances.

- We proposed a novel annotation method for constructing preference datasets. With that, we expanded the RAGTruth dataset (Niu et al., 2024) with responses derived from Llama-3-8B-Instruct (Grattafiori et al., 2024) and Qwen2.5-7B-Instruct (Team, 2024), and created the first human preference dataset tailored to the RAG scenario for handling both problematic queries and answerable ones.

- Extensive AL-driven model optimization was conducted on the constructed dataset, with results demonstrating the effectiveness of our approach.

## 2 RELATED WORK

### 2.1 ACTIVE LEARNING

Active learning (AL) is a widely adopted technique for optimizing the trade-off between annotation costs and model performance by selecting the most informative samples from large unlabeled datasets. Central to AL are three components: a labeling oracle, an unlabeled data pool, and a query strategy. Common strategies include uncertainty-based methods, which prioritize difficult samples based on prediction uncertainty (Beluch et al., 2018; Liu et al., 2021; Schröder et al., 2021; Maekawa et al., 2022; Rouzegar & Makrehchi, 2024), and diversity-based methods, which focus on selecting diverse samples to enrich datasets (Hasan et al., 2018; Sinha et al., 2019; Agarwal et al., 2020; Maekawa et al., 2022; Xie et al., 2023).

Active learning (AL) has been effectively applied across various NLP tasks, including text classification (Yan et al., 2020; Schröder et al., 2021), text summarization (Gidiotis & Tsoumakas, 2022; Tsvigun et al., 2022), and question answering (Karamcheti et al., 2021; Padmakumar & Mooney, 2021), achieving significant cost reductions and performance improvements. These approaches have demonstrated strong potential in optimizing model training efficiency and enhancing overall system performance. Despite their successes, existing methods often neglect the influence of inherent sample properties on diversity. Addressing this gap, our work introduces a novel approach for evaluating sample diversity in the RAG context by comparing similarities across different data fields.

### 2.2 ACTIVE LEARNING MEETS LLMS

As large language models (LLMs) continue to advance, their integration with AL has become a focal point for addressing high annotation costs (Tan et al., 2024) and challenges in effective knowledge utilization (Xu et al., 2024). Currently, the integration of AL with LLMs primarily involves three approaches: employing traditional active learning methods to select samples for the downstream processes of LLMs (e.g., fine-tuning, in-context learning, evaluation) (Xie et al., 2023; Margatina et al., 2023; Bayer & Reuter, 2024), utilizing LLMs to assess sample quality (e.g., uncertainty estimation) (Li et al., 2024), and leveraging LLMs to replace human annotators (Xiao et al., 2023; Kholodna et al., 2024). For instance, Margatina et al. (2023) demonstrated the effectiveness of similarity sampling for classification, framing in-context learning's example selection as a single-round AL task. Li et al. (2024) proposed LDCAL for text summarization, while Rouzegar & Makrehchi (2024) balanced cost and accuracy in text classification. Other studies addressed noisy data filtering (Taneja & Goel, 2024) and explored the use of LLMs as annotators (Zhang et al., 2023), highlighting both strengths and limitations.

Our research integrates AL into the RAG framework, leveraging its capabilities to address the unique challenges of fine-tuning LLMs. Specifically, we focus on selecting high-impact samples that enhance model performance while considering diversity within the RAG setting. To the best of our knowledge, this is the first study to explore AL-driven optimization for LLMs in the RAG context.

## 3 PROBLEM DEFINITION & PRELIMINARIES

### 3.1 PROBLEM DEFINITION

To enhance a model's performance under RAG setting, we employ preference optimization based on the model's own conversational history. The goal is twofold: to strengthen the model's ability to reject queries it cannot answer accurately and consistently, and to improve response stability for

in-capability queries. Each unlabeled sample consists of a user query $q$, a reference document $r$, and a model-generated answer $a$. We identify the most informative samples and label them by assessing consistency between $a$ and $r$, then use these curated samples for preference optimization, thereby **refining the model under limited annotation resources**.

Inspired by prior works (Tsvigun et al., 2022; Margatina et al., 2023), we adopt a diversity-based approach, evaluating diversity by computing distances among samples to prioritize varied and informative data. Experimental results in Section 6 and Appendix A also prove its effectiveness.

### 3.2 DIVERSITY-BASED ACTIVE LEARNING

Given an unlabeled dataset $U = \{x_1, x_2, \ldots, x_n\}$, feature extraction algorithms (e.g., TF-IDF) transform each sample $x_i$ into a feature vector $\mathbf{x_i}$. Initially, $k$ samples are randomly selected to form the initial selected set $S = \{s_1, s_2, \ldots, s_k\}$, and these samples are removed from $U$.

In subsequent rounds, the average distance $\mathcal{D}$ between each sample $x_i$ in $U$ and all samples in $S$ is measured as:

$$\mathcal{D}(x_i) = \frac{1}{|S|} \sum_{j=1}^{|S|} (1 - \frac{\mathbf{x_i} \cdot \mathbf{s_j}}{\|\mathbf{x_i}\| \|\mathbf{s_j}\|}) \tag{1}$$

The top-$k$ samples ranked by $\mathcal{D}$ are added to $S$ and subsequently removed from $U$. This process repeats until the annotation budget is reached.

Unlike methods that label samples during each selection round, we label all samples in $S$ collectively at the end, reducing annotators' waiting time and improving efficiency.

### 3.3 RAG FRAMEWORK

In the domain of NLP, RAG is a crucial methodology. Suppose a user inputs a natural language query $q$, there is a pre-established knowledge repository housing a collection of text chunks $r_i$. The system computes a set of relevance scores $f(q, r_i)$ to screen relevant text chunks from this repository against $q$. Next, based on a predefined threshold $\tau$ or a predefined number $k$, the text chunks that meet the condition $f(q, r_i) > \tau$ or the top-$k$ relevant chunks are retrieved and ranked, generating a set $R = \{r_1, r_2, \ldots, r_k\}$. These retrieved chunks are combined with $q$ into a prompt $P = [q; R]$, which is the input to an LLM $\mathcal{M}$ to generate an answer $a = \mathcal{M}(P)$. By leveraging the retrieved text, the generated answer attains enhanced contextual accuracy.

Evidently, in contrast to other scenarios, within the RAG framework, each conversation of the model encompasses multiple attributes, specifically the user-posed query, the retrieved references, and the answer generated by the model. In this setting, the measurement of distances between samples becomes more complicated. The direct application of AL can lead to inaccurate measurement of distances among samples. Hence, it is imperative to develop a method for measuring sample distances tailored to the RAG scenario.

## 4 AL4RAG

To train models to address both out-of-capability and in-capability queries while working within a limited annotation budget, we need to select a subset of the most informative samples from the model's conversation records. Next, we will annotate these samples and construct a preference dataset. Finally, we use these annotated samples for DPO training. The specific process is as follows:

### 4.1 ACTIVE LEARNING PROCESS

We employ AL to select informative samples from the RAG model's historical conversation records. Specifically, the process entails the following steps:

(1) Random selection of initial samples for labeling. Initially, a small subset of samples is selected randomly from the entire dataset as the selected set.

(2) Measurement of similarity between remaining and selected samples. The second step involves measuring the similarity between unselected samples and the selected ones. In this step, we use the

user input (e.g., user query, question) $q$ as the unit for measuring sample similarity.

$$\text{sim}(x, y) = \frac{\mathbf{x_q} \cdot \mathbf{y_q}}{\|\mathbf{x_q}\|\|\mathbf{y_q}\|} \tag{2}$$

where $\mathbf{x_q}, \mathbf{y_q}$ refer to queries of sample $x$ and $y$.

(3) Measurement of similarity among remaining samples. The same method is used to measure the similarity among the remaining samples in the unselected set.

(4) Scoring of remaining samples. An IDDS score (Tsvigun et al., 2022) is assigned to each of the remaining samples based on the equation as follows:

$$\text{IDDS}(x) = \lambda \frac{\sum_{j=1}^{|U|} \text{sim}(\mathbf{x}, \mathbf{x_j})}{|U|} - (1 - \lambda) \frac{\sum_{i=1}^{|S|} \text{sim}(\mathbf{x}, \mathbf{x_i})}{|S|} \tag{3}$$

where $\lambda$ is a hyper-parameter, $U$ refers to the unselected set and $S$ refers to the selected set.

(5) Expansion of the selected set: Move the top %k samples, based on their scores, to the selected set. Then, proceed back to step (2).

Our goal is to pick samples that are different from the ones already selected but similar to those not yet chosen, thereby excluding outliers. This process iterates until the target sample count is achieved. The detailed pseudo code is shown in Appendix D.

## 4.2 DATA ANNOTATION & PREFERENCE DATASET CONSTRUCTION

After obtaining an informative subset through AL, we need to annotate these samples to construct a preference dataset. In general, the construction of a preference dataset involves the annotator choosing the better one among two candidate answers $(a_1, a_2)$ to an input prompt $p$. Then, the better one is set as $a_w$, and the other one is set as $a_l$. However, in practical applications of the RAG model, users typically request only a single response from the model for a given query. As a result, we are unable to obtain a pair of answers to a query. For our task, we propose a novel method for dataset construction. Specifically, we ask annotators to assess the reliability of the model's response in each RAG conversation and then generate a label $h$, where $h = 0$ indicates that the response has no unreliable elements, and $h = 1$ indicates that the response contains unreliable element. Here, "unreliable element" refer to content in the model's response that is **unsupported by or contradict with the reference**, which is also known as faithfulness hallucination in Huang et al. (2025). Next, we examine the $h$ label of each sample. For samples where $h = 0$, we designate the model's original response as $a_w$ and an explicit rejection response as $a_l$. Conversely, for samples where $h = 1$, we assign the explicit rejection response as $a_w$ and the model's original response as $a_l$. By modifying the feedback strategy, we successfully construct a preference dataset tailored for scenarios involving only a single model response. The detailed pseudo code can be found in Appendix D.

## 4.3 FINE-TUNING PROCESS

Inspired by previous work (Khaki et al., 2024), we adopt Direct Preference Optimization (DPO) (Rafailov et al., 2024) to achieve the aforementioned objectives. DPO fine-tunes LLMs to align their outputs with human preferences, simplifying optimization by eliminating the need for complex reward function estimation (Ouyang et al., 2022; Bai et al., 2022).

With the constructed preference dataset, we define two policies: $\pi_\theta$, the model being optimized, and $\pi_o$, the original model used as a baseline. The optimization phase involves minimizing a loss function based on human preferences:

$$\mathcal{L}_{\text{DPO}}(\theta) = -\log \sigma \beta \left( \log \frac{\pi_\theta(a_w|p)}{\pi_o(a_w|p)} - \log \frac{\pi_\theta(a_l|p)}{\pi_o(a_l|p)} \right) \tag{4}$$

where $\sigma$ is the sigmoid function, and $\beta$ adjusts alignment speed with human preferences.

By fine-tuning the model with DPO, we can achieve the previously outlined optimization objectives, thereby mitigating model trust issues caused by unreliable responses.

## 5 AL4RAG$_{\mathrm{RAS}}$

### 5.1 OBSERVATION

*Conventional sample similarity measurement based on user input can lead to inaccurate judgment of sample distances.* Classical Active Learning (AL) has achieved significant success in both Computer Vision (CV) (Budd et al., 2021; Tuia et al., 2021; Wang et al., 2023) and Natural Language Processing (NLP) (Zhang et al., 2022), with numerous methods leveraging sample diversity to enhance learning efficiency (Shi et al., 2021; Kim et al., 2021; Li et al., 2022; Jin et al., 2022). In both CV and NLP tasks such as text classification (Tan et al., 2021), text summarization (Tsvigun et al., 2022), semantic parsing (Li et al., 2023), and information extraction (Duan, 2024), samples typically consist of a single attribute (e.g., input images or text), allowing straightforward similarity measurements. In contrast, the RAG scenario involves conversation records with three distinct attributes: user query, retrieved references, and LLM-generated response. Measuring similarity solely based on user input is inadequate, as unreliability in RAG applications often stems from misunderstanding references (Chen et al., 2024). Combining multiple attributes or using prompt-based similarity also introduces inaccuracies, as length variations (e.g., long references or template text dominating short queries) artificially inflate similarity between otherwise distinct samples. To address this, our approach evaluates each attribute independently, enabling more accurate similarity measurements and clearer sample differentiation, thereby better capturing the critical role of the query. As shown in Figure 2, our method consistently outperforms input-based and prompt-based similarity across varying data ratios and tasks, demonstrating its robustness and adaptability to real-world RAG applications. For detailed experimental results, please refer to Section 6.2 and Appendix A.

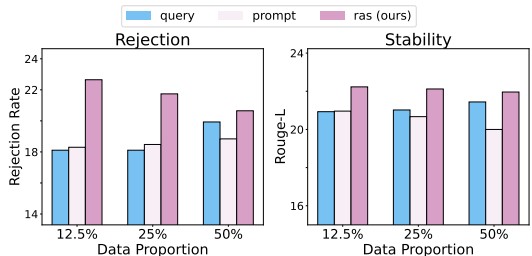

Figure 2: Part of the experimental results of query similarity, prompt similarity and *ras*, which shows the impact of different similarity measurement on model performance. The left graph shows the performance of handling out-of-capability queries, and the graph on the right shows the performance of handling model-answerable queries.

### 5.2 AL4RAG WITH RETRIEVAL-AUGMENTED SIMILARITY

Similar to the method mentioned previously, we carry out the same series of steps to screen samples. However, in step (2) and (3), to address the observed issues, we propose **retrieval-augmented similarity** (*ras*), a novel metric explicitly designed for RAG-structured data. It redefines sample comparison by holistically modeling the multi-faceted nature of RAG samples, comprising a user query $q$, retrieved reference documents $r$, and a generated answer $a$, where the prompt $p = [q; r]$ serves as the generator's input.

Unlike typical AL algorithms, which often treat the input as a single entity (Margatina et al., 2023; Taneja & Goel, 2024), *ras* adopts a structured, attribute-wise similarity mechanism. It first computes semantic similarities for the query $q$ and reference $r$ independently, then aggregates them into an intermediate value. Crucially, *ras* applies a minimum operation between this aggregate and the full-prompt similarity, effectively emphasizing the most discriminative attributes and reducing bias from imbalanced text lengths. This dual strategy ensures robustness in scenarios where lengthy references overshadow short queries or templated prompts distort the true relationships between samples. Simultaneously, it guarantees accurate measurement when queries are similar but references are not.

Formally, given two samples $x$ and $y$, the *ras* metric is defined as:

$$\mathrm{ras}(x, y) = \min\left(\frac{\mathbf{x_p} \cdot \mathbf{y_p}}{\|\mathbf{x_p}\|\|\mathbf{y_p}\|}, \frac{1}{2}\left(\frac{\mathbf{x_q} \cdot \mathbf{y_q}}{\|\mathbf{x_q}\|\|\mathbf{y_q}\|} + \frac{\mathbf{x_r} \cdot \mathbf{y_r}}{\|\mathbf{x_r}\|\|\mathbf{y_r}\|}\right)\right) \tag{5}$$

where $\mathbf{x_p}, \mathbf{y_p}, \mathbf{x_q}, \mathbf{y_q}$ and $\mathbf{x_r}, \mathbf{y_r}$ refer to $p$, $q$ and $r$ embeddings of $x$ and $y$. Detailed pseudo code is presented in Appendix D. In the main experiments of this paper, we utilized TF-IDF vectorization. In the subsequent experiments, we compared the impacts of different vectorization methods on the final performance of the model.

# 6 EXPERIMENTS

## 6.1 EXPERIMENTAL SETTINGS

**Tasks & Dataset**

We evaluated our method on Question Answering, Summary, and Data-to-text tasks using RAGTruth (Niu et al., 2024), a high-quality manually annotated dataset for RAG faithfulness, containing queries, references, model responses, and reliability labels. Since RAGTruth lacks responses from newer models, we deployed Llama-3-8B-Instruct and Qwen2.5-7B-Instruct to generate these, then used Deepseek-v3-0324 for annota-

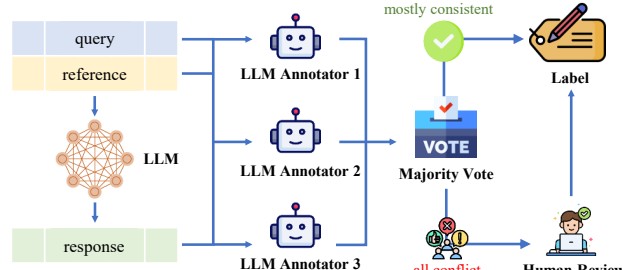

Figure 3: Quality control of LLM-assisted annotation.

tion (replacing manual work due to budget constraints). For quality control (Figure 3), each entry was annotated three times, with final labels determined by majority vote; conflicting cases (e.g., format-noncompliant outputs) were manually verified. To build the preference dataset, we selected responses from the chosen models, identified their reliability labels, and designated preferred/non-preferred answers. The resulting dataset (around 3,000 entries per model) was split into training/validation/test sets at 8:1:1, with each sample containing a query, reference, chosen answer, and rejected answer. AL strategies were applied solely to the training set.

**Model**  We selected Llama-3-8B-Instruct, Llama-2-7B-chat and Qwen2.5-7B-Instruct as our base models. However, the standard DPO process relies on an SFT model, but none exists for our task. Thus, we fine-tuned each model on its corresponding training set. This empowered the model to better evaluate query complexity and handle rejection-required scenarios.

**Baselines**  To verify the effectiveness of our method, we selected several baselines for comparison, including Random, Entropy (Wang & Shang, 2014), Coreset (Sener & Savarese, 2017), BLEUVar (Xiao et al., 2020), Naive IDDS (Tsvigun et al., 2022), LDCAL (Li et al., 2024), IDDS$_{prompt}$ and IDDS$_{q,r\ combined}$. For baseline introductions, please refer to Appendix A.1.

**Evaluation Metrics**  For the performance of refusing to answer out-of-capability queries, we use **Rejection Rate (RR)** and **F1** score for evaluation. For the performance of correctly answering queries, we measure it by the text similarity between the model's responses and the correct reference answers, including **Rouge-L** and **BERTScore**. Furthermore, we employ Spark X1 to evaluate the **Faithfulness** of model-generated answers by quantifying their consistency with reference materials. Specifically, we present the proportion of answers that demonstrate alignment with the references.

**Implementation Details**  For the AL process, we set the number of iteration rounds for both AL4RAG and IDDS to 5. In the fine-tuning processes, we employ LoRA (Hu et al., 2021) with a learning rate of $1 \times 10^{-5}$ and train for one epoch. During the generation phase, we set the temperature of the LLM to 0.7 and report the mean performance and MSE obtained from five independent generation runs. Note that MSE for Rouge-L and BERTScore are not provided. This is because their MSE values are exceedingly small, making it impossible to discern differences among the results.

## 6.2 MAIN RESULTS

In the main context, we report the results of Llama-3-8B-Instruct. For results of Llama-2-7B-chat and Qwen2.5-7B-Instruct, please refer to Appendix A.2 and A.3.

### 6.2.1 REJECTION RESULTS

Table 1 compares the performance of models trained on data selected by different active learning methods in handling out-of-capability queries, evaluated at data usage proportions of 12.5%, 25%, and 50% with Llama-3-8B-Instruct as the base model. Our method consistently outperforms all baselines across all data ratios for both models. Interestingly, as data proportion increases, our

Table 1: Overall rejection performance of different algorithms under different data proportions when applied to Llama-3. The best results are **bolded**, and the second-best results are underlined.

| Algorithms | 12.5% | | 25% | | 50% | |
|---|---|---|---|---|---|---|
| | **RR** | **F1** | **RR** | **F1** | **RR** | **F1** |
| Random | $23.04_{\pm0.03}$ | $28.25_{\pm0.06}$ | $23.04_{\pm0.01}$ | $27.60_{\pm0.05}$ | $24.02_{\pm0.24}$ | $28.13_{\pm0.42}$ |
| Entropy | $26.96_{\pm0.01}$ | $30.47_{\pm0.00}$ | $23.53_{\pm0.01}$ | $27.78_{\pm0.04}$ | $23.53_{\pm0.27}$ | $28.27_{\pm0.47}$ |
| Coreset | $26.96_{\pm0.13}$ | $30.19_{\pm0.09}$ | $25.98_{\pm0.21}$ | $31.06_{\pm0.21}$ | $22.06_{\pm0.01}$ | $25.55_{\pm0.00}$ |
| BLEUVar | $25.00_{\pm0.10}$ | $28.50_{\pm0.16}$ | $\underline{26.96}_{\pm0.08}$ | $30.85_{\pm0.06}$ | $25.49_{\pm0.06}$ | $31.09_{\pm0.12}$ |
| Naive IDDS | $25.00_{\pm0.13}$ | $30.32_{\pm0.16}$ | $22.55_{\pm0.02}$ | $26.58_{\pm0.05}$ | $23.04_{\pm0.06}$ | $26.87_{\pm0.09}$ |
| LDCAL | $23.04_{\pm0.12}$ | $27.12_{\pm0.06}$ | $25.98_{\pm0.03}$ | $\underline{31.24}_{\pm0.00}$ | $\underline{28.43}_{\pm0.06}$ | $\underline{32.14}_{\pm0.08}$ |
| IDDS$_{prompt}$ | $25.49_{\pm0.21}$ | $29.73_{\pm0.10}$ | $23.53_{\pm0.00}$ | $27.42_{\pm0.04}$ | $24.02_{\pm0.08}$ | $27.77_{\pm0.05}$ |
| IDDS$_{q,r\ combined}$ | $25.00_{\pm0.10}$ | $29.43_{\pm0.11}$ | $23.04_{\pm0.06}$ | $27.42_{\pm0.04}$ | $19.61_{\pm0.15}$ | $23.61_{\pm0.14}$ |
| AL4RAG | $\underline{27.45}_{\pm0.09}$ | $\underline{32.29}_{\pm0.04}$ | $25.98_{\pm0.03}$ | $30.52_{\pm0.00}$ | $23.53_{\pm0.04}$ | $28.81_{\pm0.04}$ |
| AL4RAG$_{ras}$ | $\mathbf{31.37}_{\pm0.13}$ | $\mathbf{35.37}_{\pm0.18}$ | $\mathbf{30.39}_{\pm0.21}$ | $\mathbf{34.20}_{\pm0.13}$ | $\mathbf{28.43}_{\pm0.00}$ | $\mathbf{33.76}_{\pm0.02}$ |

Table 2: Overall stability performance of various algorithms across different data proportions when applied to Llama-3. The best results are **bolded**, and the second-best results are underlined.

| Algorithms | 12.5% | | | 25% | | | 50% | | |
|---|---|---|---|---|---|---|---|---|---|
| | **Rouge-L** | **BERTScore** | **Faithfulness** | **Rouge-L** | **BERTScore** | **Faithfulness** | **Rouge-L** | **BERTScore** | **Faithfulness** |
| Random | $\underline{33.83}$ | $50.97$ | $74.53_{\pm0.01}$ | $33.19$ | $50.13$ | $71.97_{\pm0.00}$ | $33.31$ | $50.48$ | $74.96_{\pm0.04}$ |
| Entropy | $33.62$ | $\underline{51.09}$ | $69.05_{\pm0.00}$ | $33.23$ | $50.56$ | $74.24_{\pm0.03}$ | $33.19$ | $50.51$ | $72.45_{\pm0.06}$ |
| Coreset | $33.59$ | $50.82$ | $\underline{74.78}_{\pm0.03}$ | $33.59$ | $50.57$ | $71.03_{\pm0.05}$ | $33.25$ | $50.08$ | $74.82_{\pm0.01}$ |
| BLEUVar | $33.38$ | $50.25$ | $74.16_{\pm0.10}$ | $33.28$ | $50.27$ | $72.37_{\pm0.03}$ | $33.63$ | $\underline{50.99}$ | $74.82_{\pm0.01}$ |
| Naive IDDS | $33.74$ | $50.74$ | $73.14_{\pm0.00}$ | $32.92$ | $49.71$ | $74.74_{\pm0.03}$ | $32.64$ | $50.01$ | $73.62_{\pm0.06}$ |
| LDCAL | $33.26$ | $50.60$ | $74.54_{\pm0.05}$ | $\underline{33.67}$ | $50.44$ | $\underline{74.75}_{\pm0.01}$ | $\underline{33.71}$ | $50.56$ | $74.79_{\pm0.07}$ |
| IDDS$_{prompt}$ | $33.37$ | $50.44$ | $73.87_{\pm0.04}$ | $32.59$ | $49.33$ | $72.30_{\pm0.01}$ | $33.05$ | $50.04$ | $69.93_{\pm0.11}$ |
| IDDS$_{q,r\ combined}$ | $33.61$ | $50.66$ | $73.47_{\pm0.02}$ | $33.23$ | $50.35$ | $72.88_{\pm0.02}$ | $\mathbf{34.10}$ | $\mathbf{51.37}$ | $68.12_{\pm0.06}$ |
| AL4RAG | $33.65$ | $50.56$ | $70.56_{\pm0.02}$ | $33.60$ | $\underline{50.96}$ | $73.65_{\pm0.00}$ | $\underline{33.71}$ | $50.97$ | $73.76_{\pm0.07}$ |
| AL4RAG$_{ras}$ | $\mathbf{34.15}$ | $\mathbf{51.13}$ | $\mathbf{75.11}_{\pm0.05}$ | $\mathbf{33.73}$ | $\mathbf{50.98}$ | $\mathbf{75.33}_{\pm0.01}$ | $33.40$ | $50.36$ | $\mathbf{75.25}_{\pm0.02}$ |

method's performance slightly declines while baselines improve, suggesting our method effectively selects impactful samples under smaller proportions, reducing noise. Additionally, all AL algorithms perform well at 12.5% and outperform random selection of 50% data, indicating AL's effectiveness with small proportions by capturing useful samples. As proportion increases, AL's advantage over random selection diminishes; some are even surpassed at 50%. However, our method maintains an edge, demonstrating its viability at high proportions.

### 6.2.2 STABILITY RESULTS

Table 2 demonstrates the performance of models trained on data selected by different active learning methods, in ensuring accurate responses to questions that fall within the model's capacity. It can be seen that our method comprehensively outperforms the baseline methods when the data ratios are 12.5% and 25%, regardless of whether it is the traditional similarity metrics or the faithfulness evaluated by the LLM. When the data ratio is 50%, our method still maintains the lead in terms of faithfulness, and also demonstrates a strong performance in traditional metrics. Notably, random selection performs remarkably well across all data ratios, especially in terms of faithfulness, outperforming most AL methods. However, our method consistently surpasses random selection, which further highlights the advancement of our method. Additionally, it can be observed that there is no obvious correlation between faithfulness and traditional similarity metrics. This further highlights the need for new text quality evaluation metrics in the era of LLMs.

### 6.3 ABLATION STUDY

In the ablation experiments, we modify the terms involved in the weighted-average part (i.e., $\frac{x_q \cdot y_q}{\|x_q\| \|y_q\|}$ and $\frac{x_r \cdot y_r}{\|x_r\| \|y_r\|}$) of equation 5. Specifically, we investigated the individual and combined effects of the components (i.e., question, reference, answer). We conducted experiments under a quarter of the Llama-3-8B-Instruct training set, with the results presented in Table 3. It can be seen that including query and reference yields optimal model performance in both tasks. Specifically,

Table 3: Effect of different components on model performance. The best results are **bolded**, and the second-best results are underlined.

| Components | | | Rejection | | Stability | | |
|---|---|---|---|---|---|---|---|
| *query* | *reference* | *answer* | RR | F1 | Rouge-L | BERTScore | Faithfulness |
| ✓ | ✗ | ✗ | $27.94_{\pm0.01}$ | $31.96_{\pm0.01}$ | 32.81 | 49.53 | $72.78_{\pm0.04}$ |
| ✗ | ✓ | ✗ | $26.96_{\pm0.18}$ | $31.33_{\pm0.16}$ | 32.87 | 49.97 | $74.67_{\pm0.02}$ |
| ✗ | ✗ | ✓ | $21.57_{\pm0.21}$ | $25.08_{\pm0.15}$ | 33.18 | 50.07 | $72.12_{\pm0.02}$ |
| ✓ | ✗ | ✓ | $20.10_{\pm0.00}$ | $24.21_{\pm0.02}$ | 33.66 | 50.64 | $73.28_{\pm0.06}$ |
| ✗ | ✓ | ✓ | $24.02_{\pm0.24}$ | $28.50_{\pm0.24}$ | 32.91 | 49.95 | $74.13_{\pm0.04}$ |
| ✓ | ✓ | ✓ | $23.04_{\pm0.15}$ | $27.44_{\pm0.23}$ | 33.59 | 50.57 | $75.11_{\pm0.09}$ |
| ✓ | ✓ | ✗ | **$30.39_{\pm0.21}$** | **$34.20_{\pm0.13}$** | **33.73** | **50.98** | **$75.33_{\pm0.01}$** |

Table 4: The impact of different vectorization methods on model performance under a quarter of the data volume. The best results are **bolded**.

| Methods | Rejection | | Stability | | |
|---|---|---|---|---|---|
| | RR | F1 | Rouge-L | BERTScore | Faithfulness |
| TF-IDF | **$30.39_{\pm0.21}$** | **$34.20_{\pm0.13}$** | **33.73** | **50.98** | **$75.33_{\pm0.01}$** |
| Sentence-BERT | $23.53_{\pm0.19}$ | $27.59_{\pm0.17}$ | 32.81 | 49.93 | $72.85_{\pm0.03}$ |
| stella | $25.49_{\pm0.06}$ | $28.97_{\pm0.07}$ | 33.29 | 50.48 | $73.51_{\pm0.04}$ |

with other components identical, models using reference info consistently outperform those without in rejection tasks. Though the reference-only model does not outperform the query-only one, this does not weaken the reference's contribution to rejection performance. Additionally, query and reference both significantly impact stability, highlighting their key roles. Notably, incorporating the answer usually worsens performance, indicating interference. Thus, we advocate combining query and reference. We further explored the impact of diverse data ratios and training steps on model performance, and the results are presented in Appendix A.4 and A.5.

## 6.4 IMPACT OF DIFFERENT VECTORIZATION METHODS

To explore the influence of different vectorization methods on the model performance, we selected two pre-trained text embedding models for comparison. One is *Sentence-BERT* (Reimers, 2019), which is widely used in AL research. The other one is *stella*[1], a 1.5B model ranked highly on the MTEB (Muennighoff et al., 2022) leaderboard[2]. Table 4 shows the performance of TF-IDF and these two text embedding models with a quarter of the data on Llama-3-8B-Instruct. It can be seen that TF-IDF significantly outperforms the two pre-trained models in both tasks. This discrepancy likely stems from fundamental differences in their approach: pre-trained models focus on capturing deep semantic relationships, while TF-IDF emphasizes surface-level text form and structure. This distinction allows TF-IDF to better differentiate samples with similar meanings but varying expressions and labels, making it particularly effective at identifying fine-grained variations. Moreover, the performance of *stella* is better than that of *Sentence-BERT* in both tasks, which indicates that more advanced text embedding models can better distinguish samples, in line with intuitive expectations.

## 7 CONCLUSION

In this work, we introduce AL4RAG, the first AL framework for RAG, proposing an effective selection strategy tailored to the unique data patterns of RAG. To improve sample differentiation, we develop retrieval-augmented similarity (*ras*), enabling more accurate measurement of sample distances. Additionally, with our proposed annotation method, we expand the RAGTruth dataset and construct the first human preference dataset for RAG, allowing models to handle both problematic and answerable queries effectively. Extensive AL-driven optimization on the constructed dataset demonstrates that our approach consistently outperforms baselines, enhancing both response stability and the model's ability to reject unreliable queries. These contributions provide a strong foundation for advancing RAG-based learning and optimizing LLMs with external knowledge.

---

[1] https://huggingface.co/dunzhang/stella_en_1.5B_v5
[2] https://huggingface.co/spaces/mteb/leaderboard

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

# A  ADDITIONAL EXPERIMENTAL DETAILS

Experiments of Llama-2-7B-chat and Qwen2.5-7B-Instruct are carried out on eight NVIDIA RTX 3090 GPUs, and experiments of Llama-3-8B-Instruct are conducted on four NVIDIA RTX A6000 GPUs. In terms of stability performance, we present the results of the ROUGE family and BERTScore for Llama-2-7B-chat and present the same metric as Llama-3-8B-Intruct with faithfulness evaluated by Deepseek-v3 for Qwen2.5-7B-Instruct.

## A.1  INTRODUCTION OF THE COMPARED BASELINES

We present a brief introduction of the compared baselines below. Among these baselines, Entropy and BLEUVar are uncertainty-based, Coreset and IDDS are diversity-based, and LDCAL combines both uncertainty and diversity.

Table 5: Overall rejection performance when base model is Llama-2-7B-chat. The best results are **bolded**, and the second-best results are underlined.

| Algorithms | 12.5% | | 25% | | 50% | |
|---|---|---|---|---|---|---|
| | **RR** | **F1** | **RR** | **F1** | **RR** | **F1** |
| Random | $18.52_{\pm0.02}$ | $29.29_{\pm0.06}$ | $18.87_{\pm0.03}$ | $29.90_{\pm0.05}$ | $20.46_{\pm0.04}$ | $31.88_{\pm0.07}$ |
| Entropy | $19.40_{\pm0.01}$ | $30.38_{\pm0.02}$ | $\underline{20.99}_{\pm0.08}$ | $32.52_{\pm0.13}$ | $20.99_{\pm0.05}$ | $32.40_{\pm0.10}$ |
| Coreset | $19.05_{\pm0.04}$ | $29.80_{\pm0.07}$ | $18.17_{\pm0.00}$ | $29.30_{\pm0.00}$ | $21.19_{\pm0.02}$ | $32.90_{\pm0.03}$ |
| BLEUVar | $19.05_{\pm0.18}$ | $29.50_{\pm0.34}$ | $20.28_{\pm0.00}$ | $31.63_{\pm0.01}$ | $20.99_{\pm0.02}$ | $32.49_{\pm0.02}$ |
| Naive IDDS | $\underline{20.11}_{\pm0.01}$ | $\underline{31.74}_{\pm0.03}$ | $20.99_{\pm0.08}$ | $\underline{32.67}_{\pm0.14}$ | $20.99_{\pm0.05}$ | $32.29_{\pm0.08}$ |
| LDCAL | $16.51_{\pm0.02}$ | $27.11_{\pm0.05}$ | $17.96_{\pm0.00}$ | $28.36_{\pm0.01}$ | $16.32_{\pm0.01}$ | $27.98_{\pm0.02}$ |
| IDDS$_{prompt}$ | $19.40_{\pm0.07}$ | $30.61_{\pm0.12}$ | $18.69_{\pm0.09}$ | $29.80_{\pm0.15}$ | $20.11_{\pm0.09}$ | $31.27_{\pm0.16}$ |
| IDDS$_{q,r\ combined}$ | $19.75_{\pm0.00}$ | $31.11_{\pm0.01}$ | $20.46_{\pm0.01}$ | $32.27_{\pm0.03}$ | $18.34_{\pm0.00}$ | $29.21_{\pm0.00}$ |
| AL4RAG | $19.05_{\pm0.07}$ | $30.03_{\pm0.13}$ | $18.87_{\pm0.06}$ | $29.43_{\pm0.12}$ | $20.63_{\pm0.02}$ | $32.38_{\pm0.04}$ |
| AL4RAG$_{ras}$ | $\mathbf{23.46_{\pm0.04}}$ | $\mathbf{35.43_{\pm0.06}}$ | $\mathbf{22.93_{\pm0.06}}$ | $\mathbf{35.53_{\pm0.14}}$ | $\mathbf{21.69_{\pm0.01}}$ | $\mathbf{33.36_{\pm0.02}}$ |

Table 6: Overall stability performance when base model is Llama-2-7B-chat. The best results are **bolded**, and the second-best results are underlined.

| Algorithms | 12.5% | | | | 25% | | | | 50% | | | |
|---|---|---|---|---|---|---|---|---|---|---|---|---|
| | Rouge-L | Rouge-1 | Rouge-2 | BERTScore | Rouge-L | Rouge-1 | Rouge-2 | BERTScore | Rouge-L | Rouge-1 | Rouge-2 | BERTScore |
| Random | 20.68 | 29.02 | 14.41 | 28.34 | 20.07 | 28.38 | 13.86 | 27.66 | 20.52 | 29.14 | 14.50 | 28.36 |
| Entropy | 20.60 | 29.28 | 14.40 | 29.09 | 21.59 | 30.15 | 15.25 | 30.00 | 20.37 | 29.08 | 14.08 | 28.10 |
| Coreset | 20.30 | 28.92 | 14.18 | 28.49 | 20.78 | 29.01 | 14.50 | 28.67 | 20.69 | 29.27 | 14.63 | 28.69 |
| BLEUVar | 19.75 | 28.48 | 13.65 | 27.38 | 21.62 | 30.46 | 15.33 | 29.89 | 20.46 | 28.99 | 14.13 | 28.21 |
| Naive IDDS | 20.49 | 29.07 | 14.47 | 28.75 | 20.65 | 29.06 | 14.24 | 29.05 | 19.80 | 27.83 | 13.85 | 27.67 |
| LDCAL | 14.82 | 20.65 | 9.54 | 18.65 | 15.96 | 22.11 | 10.81 | 20.57 | 14.45 | 19.82 | 9.21 | 18.20 |
| IDDS$_{prompt}$ | 20.96 | 29.69 | 14.88 | 28.90 | 20.67 | 29.15 | 14.44 | 28.64 | 20.00 | 28.79 | 13.95 | 28.02 |
| IDDS$_{q,r\ combined}$ | 20.70 | 29.18 | 14.38 | 28.47 | 19.92 | 28.10 | 13.66 | 27.52 | 20.69 | 28.95 | 14.35 | 28.18 |
| AL4RAG | 20.93 | 29.74 | 14.57 | 29.60 | 21.02 | 29.57 | 14.61 | 29.37 | 21.44 | 30.64 | 14.81 | 29.64 |
| AL4RAG$_{ras}$ | 22.23 | 31.17 | 15.63 | 30.90 | 22.12 | 31.08 | 15.81 | 31.01 | 21.96 | 31.45 | 15.28 | 30.82 |

- **Random**: The samples for annotation are randomly selected all at once.

- **Entropy** (Wang & Shang, 2014): An AL strategy that selects samples with the highest prediction uncertainty, measured by maximum entropy, to improve model performance.

- **Coreset** (Sener & Savarese, 2017): Select a small, representative subset of data that approximates the entire dataset's distribution for efficient learning.

- **BLEUVar** (Xiao et al., 2020): Use BLEU variance to prioritize samples with high uncertainty by treating documents as points in a high-dimensional space.

- **Naive IDDS** (Tsvigun et al., 2022): Select samples dissimilar to labeled ones but similar to unlabeled ones, based on document embeddings.

- **LDCAL** (Li et al., 2024): Employ LLMs (originally GPT-3.5, we use Deepseek-v3 instead) to partition the data and select samples with a strategy that computes the average certainty gain.

- **IDDS$_{prompt}$**: Use prompt similarity, incorporating queries, references, and template text, as the sample similarity measure, with IDDS as the query strategy.

- **IDDS$_{q,r\ combined}$**: Similar to the above method, but only concatenate the query and the reference, and use the similarity of this part as the sample similarity.

## A.2 MAIN RESULTS FOR LLAMA-2-7B-CHAT

To evaluate the effectiveness of our method on less capable base models, we applied it to Llama-2-7B-chat. Table 5 and Table 6 present the performance of models trained on data selected by different active learning methods. For the ability of rejecting queries outside the model's capability, our method consistently outperforms all baselines across various data proportions. Notably, similar to Llama-3, our method achieve the best performance when the data ratio is 12.5%, again demonstrating its superiority. It should be noted that the performance of LDCAL on Llama-2 is not as good as that on Llama-3. We believe that this is because Llama-2 itself has relatively weaker capabilities,

Table 7: Overall rejection performance when base model is Qwen2.5-7B-Instruct. The best results are **bolded**, and the second-best results are underlined.

| Algorithms | 12.5% | | 25% | |
|---|---|---|---|---|
| | **RR** | **F1** | **RR** | **F1** |
| Random | $23.08_{\pm0.00}$ | $32.73_{\pm0.00}$ | $23.08_{\pm0.00}$ | $32.73_{\pm0.00}$ |
| Entropy | $23.08_{\pm0.00}$ | $32.54_{\pm0.03}$ | $24.36_{\pm0.03}$ | $\underline{35.19_{\pm0.07}}$ |
| Coreset | $25.64_{\pm0.03}$ | $35.39_{\pm0.05}$ | $23.08_{\pm0.10}$ | $32.95_{\pm0.14}$ |
| BLEUVar | $23.85_{\pm0.02}$ | $33.45_{\pm0.06}$ | $24.62_{\pm0.05}$ | $34.39_{\pm0.10}$ |
| Naive IDDS | $24.36_{\pm0.03}$ | $35.50_{\pm0.06}$ | $24.36_{\pm0.03}$ | $34.20_{\pm0.04}$ |
| LDCAL | $\underline{25.64_{\pm0.01}}$ | $\underline{36.87_{\pm0.05}}$ | $\underline{24.62_{\pm0.02}}$ | $34.57_{\pm0.05}$ |
| AL4RAG$_{ras}$ | $\mathbf{26.92_{\pm0.03}}$ | $\mathbf{37.83_{\pm0.06}}$ | $\mathbf{26.92_{\pm0.03}}$ | $\mathbf{38.18_{\pm0.07}}$ |

Table 8: Overall stability performance when base model is Qwen2.5-7B-Instruct. The best results are **bolded**, and the second-best results are underlined.

| Algorithms | 12.5% | | | 25% | | |
|---|---|---|---|---|---|---|
| | **Rouge-L** | **BERTScore** | **Faithfulness** | **Rouge-L** | **BERTScore** | **Faithfulness** |
| Random | 41.82 | 60.38 | $57.56_{\pm0.04}$ | 41.78 | 60.53 | $56.83_{\pm0.03}$ |
| Entropy | 41.92 | 60.43 | $58.18_{\pm0.03}$ | 41.73 | 60.42 | $60.40_{\pm0.02}$ |
| Coreset | 41.44 | 60.03 | $58.92_{\pm0.01}$ | 41.49 | 60.13 | $61.01_{\pm0.01}$ |
| BLEUVar | 41.55 | 60.14 | $58.55_{\pm0.03}$ | 41.56 | 60.27 | $59.66_{\pm0.05}$ |
| Naive IDDS | 41.79 | $\mathbf{60.77}$ | $58.79_{\pm0.05}$ | 41.85 | 60.52 | $60.15_{\pm0.00}$ |
| LDCAL | $\underline{41.95}$ | 60.40 | $\underline{59.90_{\pm0.04}}$ | $\mathbf{42.04}$ | $\underline{60.72}$ | $61.13_{\pm0.02}$ |
| AL4RAG$_{ras}$ | $\mathbf{42.01}$ | $\underline{60.65}$ | $\mathbf{61.01_{\pm0.03}}$ | $\underline{41.97}$ | $\mathbf{60.81}$ | $\mathbf{61.62_{\pm0.06}}$ |

and the more difficult samples selected by LDCAL will confuse the model, resulting in a deterioration of its performance. As for the ability of ensuring accurate responses, our method consistently outperforms all baselines across various data proportions. Notably, similar to the rejection task, it achieves peak performance at a data ratio of 12.5%, a trend also observed in both IDDS-variant methods. This suggests that these methods can effectively identify informative samples even with a limited amount of data, ensuring efficient model optimization. Among them, our approach exhibits the most significant advantage, further demonstrating its effectiveness in stability performance.

### A.3 MAIN RESULTS FOR QWEN2.5-7B-INSTRUCT

To verify the effectiveness of our method on other model series, we conducted experiments on Qwen2.5-7B-Instruct with data proportions to be 12.5% and 25%. Table 7 and Table 8 present the performance of models trained on data selected by different active learning methods. Our method still maintain its superiority in both rejection performance and stability performance, demonstrating its effectiveness and generalizability across different model series.

### A.4 ABLATION STUDY ON DATA RATIOS

Figure 4 shows the performances of models trained via DPO with varying data ratios. Our method is used for data selection. It can be observed that training with a small amount of data using DPO can not only strengthen the rejection performance of the SFT model but also significantly improve its accuracy in answering questions within its ability. As the amount of training data increases, the model's ability to learn to refuse to answer questions during the fine-tuning process will first strengthen and then start to decline. When the proportion of training data reaches 100%, the rejection performance of the model is already much worse than that of the SFT-only model. Regarding the performance of answering questions within the model's ability, as the proportion of data increases, the performance of the model also shows a trend of first increasing and then decreasing. This is because the data selected by our method has almost no noise when the data proportion is small, but when the data proportion increases, the noise will also increase, leading to a decline in the model's performance. Moreover, the model trained with all the data using DPO still significantly outperforms the SFT-only model, which demonstrates the excellent effectiveness of DPO in this aspect.

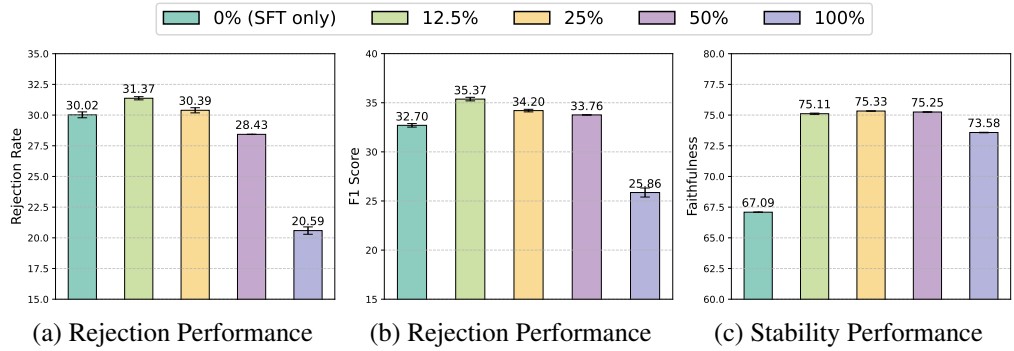

(a) Rejection Performance     (b) Rejection Performance     (c) Stability Performance

Figure 4: Ablation study on data ratios. We compared the performances of models trained via DPO with varying data ratios. (a) Rejection rate of rejection performance; (b) F1 score of rejection performance; (c) Faithfulness of stability performance.

## A.5 ABLATION STUDY ON TRAINING STEPS

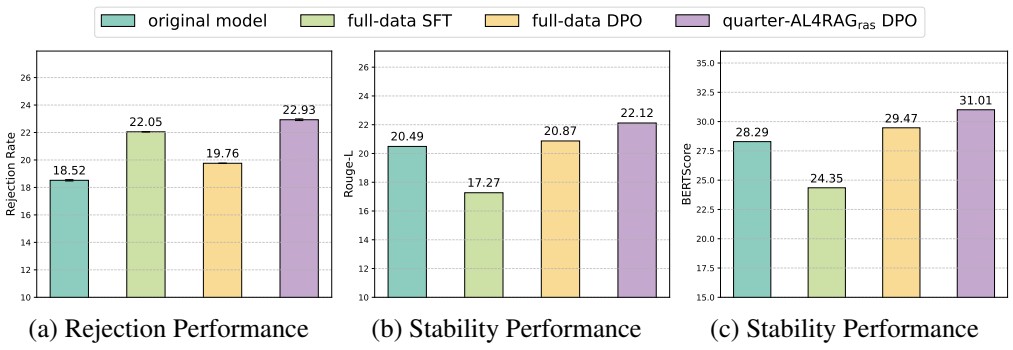

(a) Rejection Performance     (b) Stability Performance     (c) Stability Performance

Figure 5: Ablation study on training steps. We compared the performances of the original Llama-2-7B-chat model, the full-data SFT model, the full-data DPO-trained model, and the model trained via DPO with 25% data selected by our method. (a) Rejection rate of rejection performance; (b) Rouge-L of stability performance; (c) BERTScore of stability performance.

Figure 5(a) compares rejection performance across Llama-2-based models: the original model, a fine-tuned model (100% training data), a DPO-trained model (100% training data), and a DPO-trained model (25% training data, selected by our method). The model trained via DPO with data selected by our method achieves the best refusal performance, and the SFT model performs second-best, while the model trained via DPO with 100% data performs worst after the original model. This highlights that fine-tuning equips models with the ability to reject answering out-of-capability queries. Moreover, DPO training with a small amount of high-quality data further improves this capability, while training with a noisy dataset greatly weakens it. Compared to baselines, our method is the only one that consistently outperforms SFT in all data proportions, highlighting its superiority.

Figures 5(b) and 5(c) compare the stability performance of different training approaches based on Llama-2, including the original model, the SFT model trained on 100% of the training data, a DPO-trained model using 100% of the training data, and a DPO-trained model using only 25% of the data selected by our method. While fine-tuning achieves strong rejection capabilities, it significantly reduces stability compared to the original model, indicating a trade-off between the two objectives. In contrast, DPO training with 100% of the data reduces rejection performance compared to the SFT model but significantly improves stability, suggesting that optimizing for preference learning helps maintain correct answers. Furthermore, DPO training with 25% data selected by our method improves stability without compromising rejection performance, demonstrating that our approach effectively balances both goals and enhances the model's overall robustness.

## B   IMPACT STATEMENT

Our work proposes a method for efficiently selecting data samples for model training, which can make more effective use of computing resources and further contribute to sustainable development. However, there may be potential social consequences of our work. In the context of data privacy, the use of model conversation records is a critical concern. As we rely on these records to train and optimize the model, there is a potential risk of exposing sensitive user information. If proper security measures are not in place, this could lead to privacy breaches, where personal details, opinions, or interactions captured in the conversations might be accessed without authorization.

## C   PROMPTS

**Prompt for Annotating Faithfulness**

This is the "question", "reference" and "generated_answer" fields in my data. "generated_answer" is the content that the model answers the "question" according to the "reference". Your task is to check this content and determine whether "generated_answer" correctly answers the "question" with reference to the "reference". If so, output "1"; if not, output "0". Do not output any other content.
question: {question}
reference: {reference}
generated_answer: {answer}
Output:

**Prompts for Answer Generation**

**Question Answering**

Below is a question:
{question}
Below are related passages:
{reference}
Your task is to answer the question strictly based on the related passages.
In case the passages do not contain the necessary information to answer the question, please reply with: "Unable to answer based on given passages."
If you cannot answer the question precisely, please reply with: "Sorry, this question is beyond my ability."
Output:

**Summarization**

Below are some news:
{reference}
Your task is to write a summary of the news.
If you cannot summarize the news precisely, please reply with: "Sorry, this question is beyond my ability."
Output:

**Data-to-text Writing**

Your task is to write an objective overview about the following local business based only on the provided structured data in the JSON format.
You should include details and cover the information mentioned in the customers' review. The overview should be 100 - 200 words. Don't make up information.
If you cannot summarize the data precisely, please reply with: "Sorry, this question is beyond my ability."
Below are the structured data:
{reference}
Output:

# D ALGORITHMS OF AL4RAG

---

**Algorithm 1** AL4RAG: Sample Selection Algorithm for RAG Context

---

**Require:** The entire dataset $D$, user input $q$, hyper-parameter $\lambda$, the proportion $k$ of samples to be expanded each time, the expected number of samples in the selected set $N$

**Ensure:** The selected sample set $S$ after step-by-step expansion

 1: Initialize the selected sample set $S = \varnothing$
 2: Randomly select a small subset $S'$ from the dataset $D$ as the initial selected sample set
 3: $S = S'$
 4: $D = D - S'$            $\triangleright$ Remove the selected samples from the dataset
 5: **while** $D$ is not empty and $|S| < N$ **do**    $\triangleright$ Step (2): Measure the similarity between unselected samples and selected samples
 6:      **for** each unselected sample $x$ in $D$ **do**
 7:          **for** each selected sample $y$ in $S$ **do**
 8:             $\mathbf{x}_q \leftarrow$ Obtain the query representation of sample $x$
 9:             $\mathbf{y}_q \leftarrow$ Obtain the query representation of sample $y$
10:             $\text{sim}(x, y) \leftarrow \frac{\mathbf{x}_q \cdot \mathbf{y}_q}{\|\mathbf{x}_q\|\|\mathbf{y}_q\|}$
11:          **end for**
12:      **end for**          $\triangleright$ Step (3): Measure the similarity among unselected samples
13:      **for** each unselected sample $x$ in $D$ **do**
14:          **for** each unselected sample $z$ in $D$ where $z \neq x$ **do**
15:             $\mathbf{x}_q \leftarrow$ Obtain the query representation of sample $x$
16:             $\mathbf{z}_q \leftarrow$ Obtain the query representation of sample $z$
17:             $\text{sim}(x, z) \leftarrow \frac{\mathbf{x}_q \cdot \mathbf{z}_q}{\|\mathbf{x}_q\|\|\mathbf{z}_q\|}$
18:          **end for**
19:      **end for**          $\triangleright$ Step (4): Score the remaining unselected samples
20:      **for** each unselected sample $x$ in $D$ **do**
21:          $\text{sum\_sim}_U \leftarrow 0$
22:          **for** each unselected sample $x_j$ in $D$ **do**
23:             $\text{sum\_sim}_U \leftarrow \text{sum\_sim}_U + \text{sim}(x, x_j)$
24:          **end for**
25:          $\text{sim}_U \leftarrow \frac{\text{sum\_sim}_U}{|D|}$
26:          $\text{sum\_sim}_S \leftarrow 0$
27:          **for** each selected sample $x_i$ in $S$ **do**
28:             $\text{sum\_sim}_S \leftarrow \text{sum\_sim}_S + \text{sim}(x, x_i)$
29:          **end for**
30:          $\text{sim}_S \leftarrow \frac{\text{sum\_sim}_S}{|S|}$
31:          $\text{IDDS}(x) \leftarrow \lambda \cdot \text{sim}_U - (1 - \lambda) \cdot \text{sim}_S$
32:      **end for**          $\triangleright$ Step (5): Expand the selected sample set
33:      Sort the samples in $D$ in descending order according to their IDDS scores
34:      Select the top $k\%$ of the samples, denoted as set $E$
35:      $S \leftarrow S \cup E$
36:      $D \leftarrow D - E$
37: **end while**
38: **return** $S$

---

---

**Algorithm 2** Preference Dataset Construction for Single-Response RAG Model

---

**Require:** A set of samples obtained through Active Learning (AL), a RAG model, annotators
**Ensure:** A preference dataset $P$
 1: Initialize the preference dataset $P = \varnothing$
 2: **for** each sample $s$ in the set of samples obtained through AL **do**
 3:     $p \leftarrow s.prompt$                                    ▷ Get the input prompt $p$ from sample $s$
 4:     $r \leftarrow s.response$                               ▷ Get the model's response $r$ from sample $s$
 5:     $h \leftarrow$ annotators.assessReliability($r$)                ▷ Assess Reliability labels
 6:     $a_{reject} \leftarrow$ generateExplicitRejectionResponse()    ▷ Generate an explicit rejection response
 7:     **if** $h = 0$ **then**
 8:         $a_w \leftarrow r, a_l \leftarrow a_{reject}$
 9:     **else**
10:         $a_w \leftarrow a_{reject}, a_l \leftarrow r$
11:     **end if**
12:     $P \leftarrow P \cup \{(p, a_w, a_l)\}$               ▷ Add the pair $(p, a_w, a_l)$ to the preference dataset
13: **end for**
14: **return** $P$

---

**Algorithm 3** Sample Screening with Retrieval-Augmented Similarity (RAS)

---

**Require:** The entire dataset $D$, hyper-parameter $\lambda$, the proportion $k$ of samples to be expanded each time, the expected number of samples in the selected set $N$
**Ensure:** The selected sample set $S$ after step-by-step expansion
 1: Initialization                     ▷ The initialization steps are the same as those of Algorithm 1.
 2: **while** $D$ is not empty and $|S| < N$ **do**   ▷ Step (2): Measure the retrieval-augmented similarity between unselected samples and selected samples
 3:     **for** each unselected sample $x$ in $D$ **do**
 4:         $\mathbf{x_q} \leftarrow$ Obtain the query representation of sample $x$
 5:         $\mathbf{x_r} \leftarrow$ Obtain the reference representation of sample $x$
 6:         $\mathbf{x_p} \leftarrow$ Combine $\mathbf{x_q}$ and $\mathbf{x_r}$ to form the prompt representation of sample $x$
 7:         **for** each selected sample $y$ in $S$ **do**
 8:             $\mathbf{y_q} \leftarrow$ Obtain the query representation of sample $y$
 9:             $\mathbf{y_r} \leftarrow$ Obtain the reference representation of sample $y$
10:             $\mathbf{y_p} \leftarrow$ Combine $\mathbf{y_q}$ and $\mathbf{y_r}$ to form the prompt representation of sample $y$
11:             $\text{sim}_q \leftarrow \frac{\mathbf{x_q} \cdot \mathbf{y_q}}{\|\mathbf{x_q}\|\|\mathbf{y_q}\|}$
12:             $\text{sim}_r \leftarrow \frac{\mathbf{x_r} \cdot \mathbf{y_r}}{\|\mathbf{x_r}\|\|\mathbf{y_r}\|}$
13:             $\text{sim}_p \leftarrow \frac{\mathbf{x_p} \cdot \mathbf{y_p}}{\|\mathbf{x_p}\|\|\mathbf{y_p}\|}$
14:             $\text{ras}(x, y) \leftarrow \min(\text{sim}_p, \frac{1}{2}(\text{sim}_q + \text{sim}_r))$
15:         **end for**
16:     **end for** ▷ Step (3): Measure the retrieval - augmented similarity among unselected samples
17:     **for** each unselected sample $x$ in $D$ **do**
18:         Obtain $\mathbf{x_q}, \mathbf{x_r}, \mathbf{x_p}$        ▷ The same representation obtainment for sample $x$ as Step (2).
19:         **for** each unselected sample $z$ in $D$ where $z \neq x$ **do**
20:             $\mathbf{z_q} \leftarrow$ Obtain the query representation of sample $z$
21:             $\mathbf{z_r} \leftarrow$ Obtain the reference representation of sample $z$
22:             $\mathbf{z_p} \leftarrow$ Combine $\mathbf{z_q}$ and $\mathbf{z_r}$ to form the prompt representation of sample $z$
23:             $\text{sim}_q \leftarrow \frac{\mathbf{x_q} \cdot \mathbf{z_q}}{\|\mathbf{x_q}\|\|\mathbf{z_q}\|}$
24:             $\text{sim}_r \leftarrow \frac{\mathbf{x_r} \cdot \mathbf{z_r}}{\|\mathbf{x_r}\|\|\mathbf{z_r}\|}$
25:             $\text{sim}_p \leftarrow \frac{\mathbf{x_p} \cdot \mathbf{z_p}}{\|\mathbf{x_p}\|\|\mathbf{z_p}\|}$
26:             $\text{ras}(x, z) \leftarrow \min(\text{sim}_p, \frac{1}{2}(\text{sim}_q + \text{sim}_r))$
27:         **end for**
28:     **end for**
29:     Subsequent steps     ▷ The subsequent steps are the same as those of Algorithm 1, including scoring and expanding the set.
30: **end while**
31: **return** $S$

