# OpenReview forum: "Enhancing RAG with Active Learning on Conversation Records: Reject Incapables and Answer Capables"
_ICLR.cc/2026/Conference — ICLR 2026 Conference Withdrawn Submission_

### Official Review · Reviewer_PnkH · 2025-10-22

**Soundness:** 1
**Presentation:** 2
**Contribution:** 2
**Rating:** 2
**Confidence:** 4

**Summary:**

This paper introduces AL4RAG, the first Active Learning framework designed to address unreliability in Retrieval-Augmented Generation (RAG) models. Given that high-quality annotated data is scarce, AL4RAG strategically selects the most informative samples from existing LLM conversations for expert annotation, optimizing limited budgets. A key innovation is a novel metric, retrieval-augmented similarity, which is tailored for RAG's unique data patterns, unlike traditional active learning metrics. This method was used to expand the RAGTruth dataset and create the first human preference dataset for RAG. Experiments show AL4RAG outperforms baselines, improving response stability and enabling the model to effectively reject unreliable queries.

**Strengths:**

This paper has the following strengths:

- Its primary strength is its novelty in proposing AL4RAG, which appears to be the first Active Learning framework specifically tailored for the RAG context. The introduction of a selection strategy designed for the unique data patterns of RAG is a valuable contribution.
- A significant practical contribution is the new annotation method and the resulting datasets. The expansion of the RAGTruth dataset and the creation of the first human preference dataset for RAG are valuable resources for the community.
- It is supported by extensive experiments that demonstrate the proposed method's superiority. The fact that AL4RAG consistently outperforms baselines across multiple metrics provides strong validation for the framework's design and effectiveness.

**Weaknesses:**

This paper has the following weaknesses:

- The paper's core premise for using active learning – the high cost and scarcity of human annotation – is significantly undermined by its own experimental methodology. The authors use a powerful LLM (Deepseek-v3-0324) as a substitute for human annotators. This raises a critical question: If a powerful LLM is available, why is an active learning selection framework necessary? The paper fails to justify why one wouldn't simply use a powerful LLM to pseudo-label a much larger dataset, bypassing the need for selective sampling. This apparent contradiction weakens the argument for the framework's practical utility.
- Directly related to the first point, the paper provides no evaluation of the annotation quality from Deepseek-v3-032. Since this LLM serves as the "ground truth" for the experiments, its reliability is paramount. Without a human-validated study of the LLM's accuracy, the trustworthiness of the entire experimental result set is called into question.
- The design of the core retrieval-augmented similarity metric in Equation (5) is not well-justified. Specifically, the rationale for using the min operator between the two similarity terms is unclear. An ablation study is critically needed to demonstrate the benefit of this specific formulation. For example, how does the model perform if only the prompt similarity (the first term) is used?
- Key terms are left undefined, hindering reproducibility and clarity. For instance, the term “explicit rejection response” (Section 4.2) is central to the annotation process but is never clearly defined. It should provide a precise definition.
- Since a primary motivation for active learning is efficiency in low-data regimes, the experiments on data proportion (Table 1) should include results for smaller fractions, such as 5%. This would provide a stronger test of the framework's sample efficiency.
- The evaluation of embedding models (Table 4) used for the similarity metric feels dated. To provide a more robust benchmark, it should include comparisons against current state-of-the-art models, such as OpenAI's text-embedding-3-small or other top-performing models from the MTEB leaderboard.

**Questions:**

1. The paper motivates Active Learning by the scarcity of human annotation, yet substitutes an LLM (Deepseek-v3-0324) for human annotators in the experiments. This raises a key question: what is the practical utility of AL4RAG if a capable LLM is already available to pseudo-label the entire dataset, bypassing the need for sample selection?
2. Given that Deepseek-v3-032 serves as the "ground truth" annotator for the experiments, was any human validation performed to assess the quality and reliability of its annotations? Without this, the trustworthiness of the experimental results is unclear.
3. The design of the retrieval-augmented similarity metric in Equation (5) is not well-justified. Could the authors provide the rationale for using the min operator and include an ablation study comparing this formulation to simpler alternatives (e.g., using only the prompt similarity term)?
4. The term "explicit rejection response" is central to the annotation method in Section 4.2 but is never clearly defined. Could the authors provide a precise definition and examples to ensure clarity and reproducibility?
5. Given that a primary benefit of Active Learning is sample efficiency, would it be possible to include results for a smaller data proportion, such as 5%, in Table 1 to better demonstrate the framework's effectiveness in low-data regimes?
6. The embedding model evaluation in Table 4 does not include current state-of-the-art baselines. How would the retrieval-augmented similarity metric's performance compare against more recent and powerful models, such as OpenAI's text-embedding-3-small?

---

### Official Review · Reviewer_HVp7 · 2025-10-31

**Soundness:** 3
**Presentation:** 3
**Contribution:** 3
**Rating:** 4
**Confidence:** 4

**Summary:**

This work presents the novel application of active learning to retrieval-augmented generation. In addition, it presents a novel distance metric that allows for a diverse query, where standard distance metrics in text-based data often underperform. The combined method significantly outperforms AL baselines on various performance metrics and models.

**Strengths:**

- The proposed method has been evaluated with different models, leading to robust results.
- The approach seems novel but intuitively explained, making it easy to follow.
- The appendix provides further ablations, deepening the investigations into the method.
- Following the claims of the authors, the combination of AL and RAG is novel, and the proposed distance metric is as well.

**Weaknesses:**

- Only one dataset was used for the AL setting, and it is quite small (3000 samples).
- Some of the experimental settings are unclear, e.g., including the number of samples queried in each selection, starting budget, etc..
- The presented performance metrics seem to be points in time, e.g., 12.5\% of data, 25\% of data, .... For AL, providing the AUC values is crucial, as it demonstrates consistent performance throughout the AL process. Additionally, "learning curves" could be provided, which is another great indicator for comparing different methods.
- Newer diversity methods like TypiClust (Hacohen et al. (2022)) and MaxHerding (Bae et al. (2024)) should be included, also combined with the proposed distance metric for possible improvements.
- Using topk to select scored samples comes with the downside of mutual information, as it can select two samples that are both highly important but close by. Here again, using TypiClust or the GCoverage from MaxHerding could further improve this work.

**Questions:**

- Could you provide classic reports of AL metrics, including learning curves and AUC-summarized metrics?
- How do you justify the topk-selection for your strategy, and have you considered alternatives?
- Are there further datasets you could include to strengthen your empirical evidence?
- Is it possible to include TypiClust and MaxHerding with standard distance metrics as well as ras to your experimental section?

---

### Official Review · Reviewer_V2Q6 · 2025-10-31

**Soundness:** 3
**Presentation:** 3
**Contribution:** 2
**Rating:** 4
**Confidence:** 4

**Summary:**

This paper proposes AL4RAG, a framework that integrates Active Learning (AL) with Retrieval-Augmented Generation (RAG) to enhance the reliability of Large Language Models (LLMs). The core idea is to use a diversity-based AL strategy to select the most informative samples from unlabeled RAG conversation logs. These samples are then annotated to construct a preference dataset, which is used to fine-tune the base model via Direct Preference Optimization (DPO). A key contribution is the proposed Retrieval-Augmented Similarity (RAS) metric, which better captures sample similarity in the multi-attribute RAG context compared to query-only or prompt-based methods. Experiments on multiple models and tasks demonstrate improved performance in both rejecting out-of-capability queries and answering in-capability ones stably.

**Strengths:**

1. The paper accurately identifies a critical pain point in RAG systems, *i.e.*, the model's inability to reliably refuse queries beyond its capability, and tackles it by introducing AL to annotate a small number of samples for fine-tuning, thereby equipping the model with the know-when-to-refuse ability.
2. The proposed Retrieval-Augmented Similarity (RAS) effectively handles the multi-component nature of RAG data, mitigating the bias inherent in traditional query-only or prompt-based similarity measures.
3. The authors conducted comprehensive validation across multiple mainstream open-source models, using different data utilization ratios and comparing against various AL baselines. The separate evaluation of the two core objectives, *i.e.*, "refusing to answer" and "correctly answering",  makes the results analysis more reasonable and convincing.
4. Extending the RAGTruth dataset with new model outputs and constructing a RAG-specific preference dataset is a valuable contribution to the community.

**Weaknesses:**

1. The core framework of AL sampling and DPO fine-tuning builds on established techniques. The primary contribution, the RAS metric, is essentially an engineering-focused feature fusion technique, *i.e.*, combining and taking the minimum of query, reference, and prompt similarities. It lacks profound theoretical insight or mechanistic innovation.
2. The examples used for evaluating rejection capability is essentially the same standard used for constructing the preference dataset. This raises the concern that the model might merely be learning the annotator's specific refusal criteria rather than genuinely acquiring a principled, fact-based self-awareness. The extent to which the performance improvement stems from true generalization versus overfitting to a particular annotation schema requires further substantiation, for instance, by testing model transferability to new tasks.
3. Metrics like BERTScore and ROUGE-L, used to evaluate the performance of correctly answering queries, measure similarity to a reference answer. However, in RAG, a high-quality answer might involve summarizing, refining, or paraphrasing the reference information, which these metrics can penalize despite being correct. This potentially undermines the assessment of the answer's factual accuracy. Employing an LLM-as-a-judge mechanism could complement the evaluation.
4. The idea that AL selects the most informative samples to effectively teach the model is not empirically supported. What specific characteristics make the selected samples informative, for example? Are they queries that are more complex or references that are more ambiguous? A deeper analysis of the properties of the selected samples is lacking.
5. The method involves several key hyperparameters (*e.g.*, *λ* in the IDDS score). An analysis of their sensitivity and impact on final performance is missing, which is important for understanding the method's robustness and for reproducibility.

**Questions:**

1. Regarding Figure 3, do "LLM Annotator 1, 2, 3" represent three different LLM annotators, or three runs of the same LLM annotator (Deepseek-v3-0324)? If it's the same LLM, how was variation introduced (*e.g.*, via temperature)? What proportion of cases required manual review? If the three runs agreed but were all incorrect, was there a subsequent manual review stage, or were these erroneous samples included in the final dataset?
2. The primary reasons a query is unanswerable in RAG are often insufficient information in the reference documents or the model's failure to correctly extract information. Should the RAS metric focus more on capturing similarity in terms of "reference document adequacy" or "QA pair difficulty" rather than simple textual semantic similarity? How do authors justify that the current embedding-based similarity calculation for query, reference, and prompt is the optimal way to find the most informative samples?
3. The exact content of the rejection response is fixed. Are authors concerned that DPO training might cause the model to overfit to this specific phrase, rather than learning to express refusal semantically?
4. Could authors provide a qualitative or quantitative analysis of the samples selected by AL versus random selection? For instance, are they more diverse? Do they contain more questions that seem answerable but are not? Without such analysis, how can we be confident that AL4RAG's success is attributable to its selection mechanism rather than chance?

---

### Official Review · Reviewer_BSLm · 2025-11-08

**Soundness:** 2
**Presentation:** 2
**Contribution:** 2
**Rating:** 4
**Confidence:** 4

**Summary:**

This paper proposes AL4RAG, an active learning framework tailored for RAG systems. The authors introduce a novel similarity metric, retrieval-augmented similarity, to better measure sample distances in RAG's multi-attribute data structure (query, reference, response). Using this, they apply diversity-based active learning to select informative samples from model conversation logs, annotate them via a custom protocol to construct a preference dataset, and fine-tune LLMs using DPO. Experiments on multiple base models (Llama-3-8B, Llama-2-7B, Qwen2.5-7B) and tasks (QA, summarization, data-to-text) show consistent improvements in rejection of out-of-capability queries and stability on in-capability queries over several baselines, especially at low data budgets.

**Strengths:**

This paper addresses a meaningful and timely challenge in RAG systems, i.e., reducing unreliable outputs and improving selective question-answering. The proposed ras metric seems to be thoughtful adaptation to RAG’s multi-attribute structure, and the method for constructing a single-response preference dataset is clever.

It is an interesting method to utilize the idea of active learning to enable the model to reject the answer.

**Weaknesses:**

My major concern lies in the technical contribution of this paper. While the ras metric is tailored to RAG, it is a straightforward combination of existing similarity measures with a min operation. The overall AL framework builds heavily on prior diversity-based methods (e.g., IDDS), with incremental rather than foundational contributions.

Besides, there are some issues about the presentation. For example, the paper may over-claim the performance gains: the improvements over baselines, while consistent, are often marginal (e.g., 1–4% in F1 or faithfulness). The claim of "significant" improvement is overstated.
Also, the claim of being the "first AL framework for RAG" is bold but not sufficiently contextualized against related work like ActiveRAG [1].

There is also a small point: it's very hard to follow the main contribution of this paper. For example, the first paragraph discusses a lot about the RAG, while the proposed method tries to teach the model to reject the answer. The main contribution should be highlighted and more consistent across the paper.

## Reference

[1] ActiveRAG: Revealing the Treasures of Knowledge via Active Learning. Xu, et al.

**Questions:**

Please refer to the weakness part.

---

### Note · Authors · 2025-11-22

I have read and agree with the venue's withdrawal policy on behalf of myself and my co-authors.